# Unraveling the impact of congenital deafness on individual brain organization

Lenia Amaral[1†], Xiaosha Wang[2,3†], Yanchao Bi[2,3,4*], Ella Striem-Amit[1*]

[1]Department of Neuroscience, Georgetown University Medical Center, Washington DC, United States; [2]State Key Laboratory of Cognitive Neuroscience and Learning and IDG/McGovern Institute for Brain Research, Beijing Normal University, Beijing, China; [3]Beijing Key Laboratory of Brain Imaging and Connectomics, Beijing Normal University, Beijing, China; [4]Chinese Institute for Brain Research, Beijing, China

**\*For correspondence:**
ybi@bnu.edu.cn (YB);
striemit@gmail.com (ES-A)

[†]These authors contributed equally to this work

## eLife Assessment

This study presents **valuable** data on the increase in individual differences in functional connectivity with the auditory cortex in individuals with congenital/early-onset hearing loss compared to individuals with normal hearing. The evidence supporting the study's claims is **convincing**, although additional work using resting-state functional connectivity in the future could further strengthen the results. The work will be of interest to neuroscientists working on brain plasticity and may have implications for the design of interventions and compensatory strategies.

**Abstract** Research on brain plasticity, particularly in the context of deafness, consistently emphasizes the reorganization of the auditory cortex. But to what extent do all individuals with deafness show the same level of reorganization? To address this question, we examined the individual differences in functional connectivity (FC) from the deprived auditory cortex. Our findings demonstrate remarkable differentiation between individuals deriving from the absence of shared auditory experiences, resulting in heightened FC variability among deaf individuals, compared to more consistent FC in the hearing group. Notably, connectivity to language regions becomes more diverse across individuals with deafness. This does not stem from delayed language acquisition; it is found in deaf native signers, who are exposed to natural language since birth. However, comparing FC diversity between deaf native signers and deaf delayed signers, who were deprived of language in early development, we show that language experience also impacts individual differences, although to a more moderate extent. Overall, our research points out the intricate interplay between brain plasticity and individual differences, shedding light on the diverse ways reorganization manifests among individuals. It joins findings of increased connectivity diversity in blindness and highlights the importance of considering individual differences in personalized rehabilitation for sensory loss.

## Introduction

Neural plasticity, a fundamental property of the brain, refers to its ability to adapt and reorganize in response to sensory input and environmental demands. Meaningful plasticity is found in response to extreme environmental scenarios, such as missing the typical input to an entire sensory channel. Extensive research into neural plasticity in congenital deafness has shown that deafness induces neural reorganization (e.g. *Allen et al., 2013*; *Almeida et al., 2015*; *Almeida et al., 2018*; *Amaral et al., 2016*; *Finney et al., 2003*; *Lomber et al., 2010*; *Ruttorf et al., 2023*; *Scott et al., 2014*; for a review, see *Alencar et al., 2019*; *Lomber et al., 2020*). For instance, the auditory cortex (AC) in deafness becomes highly responsive to visual stimuli, reflecting a compensatory adaptation to sensory loss (e.g.

**eLife digest** Our brains have an amazing ability to adapt to changes in our environment or bodies, sometimes even 'rewiring' themselves as a result. This 'rewiring' is called plasticity, and it is especially important when one or more of our senses – such as sight or hearing – do not work.

In people born with deafness, a condition termed 'congenital deafness', the part of the brain that normally processes sounds (called the auditory cortex) reorganizes itself to process information from other senses, especially vision. This cross-modal plasticity lets the auditory cortex compensate for the missing sense of hearing by forming new connections to different brain areas. However, it is still unclear if this reorganization of the auditory cortex differs across individuals born with deafness.

Amaral, Wang et al. wanted to investigate if different people's brains have distinct ways of adapting to deafness. Specifically, they tested if congenital deafness influenced the way the auditory cortex in different people was connected to other parts of the brain. They also tested if exposure to sign language early in life affected those connections.

To do this, Amaral, Wang et al. used a brain imaging technique called fMRI to scan the brains of both congenitally deaf participants and people with healthy hearing. This showed that most of the hearing participants had similar connections between the auditory cortex and other parts of the brain.

In contrast, the connectivity of the auditory cortex – particularly to brain areas that process language – was much more diverse across deaf individuals. This diversity was even present in 'native signer' deaf participants exposed to sign language very early in life. However, comparing the native signers to deaf individuals who learned sign language much later showed that the native signers had much more consistent connections between the auditory cortex and two specific areas associated with sign language comprehension. These results indicate that both deafness and early exposure to language can shape individual differences in brain plasticity.

These findings shed new light on how sensory loss and language exposure can shape people's brains in different ways. In the future, Amaral, Wang et al. hope that the knowledge gained will benefit people with deafness or hearing loss, for example, by helping develop better tools to restore hearing or contributing to more personalised approaches to language and education in general.

*Codina et al., 2017*; *Hauthal et al., 2013*; *Simon et al., 2020*). Importantly, the reorganization of the AC in deaf individuals also plays a role in language processing, responding to sign language, which uses the visual rather than the auditory modality (*Nishimura et al., 1999*; *Trumpp and Kiefer, 2018*). Although most findings in congenital deafness that suggest visual processing in the AC are caused by hearing loss, as opposed to using sign language (*Cardin et al., 2013*; *Cardin et al., 2016*; *Fine et al., 2005*), sign language itself also affects cross-modal plasticity, e.g., in the processing of motion (*Bavelier et al., 2001*; *Codina et al., 2017*; *McCullough et al., 2012*). Therefore, both hearing loss and compensatory capacities are important factors when seeking to comprehend the plastic alterations in the AC in deafness.

Overall, hearing loss promotes cross-modal plasticity in the AC and beyond it; but do all individuals with deafness undergo the same *level* or even *type* of reorganization? Or can reorganization affect deaf people differently, shedding light on the nature of plasticity at the individual level? Recent evidence on blindness suggests that the variability between individuals may even be further increased due to sensory loss (*Sen et al., 2022*). In this study, we showed that people who were congenitally blind have significantly more individual differences in brain connectivity from their deprived visual cortex beyond what is found in sighted controls. This was especially true in areas where connectivity is reshaped by blindness (*Sen et al., 2022*). This suggests that plasticity may be more variable among people than previously thought. Further, it illustrates the role of postnatal experience, in driving individual differences in brain development. Is the expansion of individual differences due to plasticity a general principle of brain development? If so, we can expect to find increased individual differences in deafness as well.

Testing this question in deafness opens an additional question. Deafness is frequently accompanied by a secondary deprivation. Deaf children born to hearing parents who are raised without direct contact to other deaf adults often suffer from delayed language acquisition as they cannot perceive spoken or sign language in their environment (*Hall, 2017*; *Mayberry et al., 2002*; *Mayberry and*

*Eichen, 1991*). This early-onset deprivation has unique effects on brain organization as well (*Cheng et al., 2023*; *Lyness et al., 2013*; *Twomey et al., 2020*; *Wang et al., 2023*). Therefore, testing individual differences in deafness allows testing a secondary question: If the absence of experience increases individual variability, would language acquisition delay cause additional variation in the link between the auditory and language systems? Or does only early-onset and long-lasting input loss cause such diversification?

Last, individual variability in neural plasticity may also impact the restoration of hearing. In terms of auditory recovery, hearing aids and cochlear implantation are the main options in auditory rehabilitation. In congenital hearing loss, cochlear implants should be applied in younger rather than older children, as the efficacy of cochlear implants decreases if implemented later (*Karltorp et al., 2020*; *Kral and Sharma, 2012*; *Lyness et al., 2013*; *Purcell et al., 2021*; *Sharma and Campbell, 2011*). However, even then, the success of their application might be dependent on the level of the reorganization of the AC: an early work showed that in children prior to cochlear implantation, the level of metabolism in their cortex, including the AC, predicted their speech perception outcomes (*Lee et al., 2001*), suggesting a challenge posed by reorganization to intact sensory restoration. In contrast, more recently, it was shown that recruitment of the broad AC (including language areas) for visual speech in deaf adults positively correlates with auditory speech perception following implantation (*Anderson et al., 2017*). Therefore, understanding the nuances of brain reorganization and specifically how it may vary among deaf individuals may enable the implementation of more effective and individualized auditory rehabilitative interventions.

Therefore, the goal of the current study is to use brain connectivity to test if individual variability is modulated by sensory loss in deafness, and how it may be affected by delayed language acquisition. We examine whether the reorganization of the AC in congenital deafness results in connectivity that is particularly variable across individuals. We predict that higher variability will be observed in deafness, indicating a significant influence of postnatal sensory loss on brain organization across sensory systems. Alternatively, if increased individual variability is not observed for the deaf, this would challenge previous findings from the blind (*Sen et al., 2022*), arguing against the idea that sensory loss promotes individual variation in general, and suggesting instead that different sensory systems may promote more consistent or variable plasticity patterns. Last, testing the role of delayed language acquisition, we predict that deaf individuals with additional delayed language acquisition may show an additional increase in their individual connectivity differences, signifying that delayed language acquisition, as a form of short-term deprivation, can also affect brain variability across individuals.

## Results

### Does AC-FC variability differ between congenitally deaf and hearing individuals?

We first investigated whether deafness causes changes to the individual differences in functional connectivity (FC) from the AC. To achieve this, FC maps were assessed within each group, the deaf and hearing groups, for their voxel-wise variability across individuals. This was accomplished through the implementation of a whole-brain voxel-level test for homogeneity of variance (Brown-Forsythe test, see Methods). We found that multiple regions showed significant intersubject variability differences in FC between the deaf and hearing groups (*Figure 1A*; see also *Supplementary file 1, table S1* for the peaks of this effect). These included areas of the left temporal lobe (superior temporal gyrus [STG] and middle temporal gyrus [MTG], including the auditory association cortex), the bilateral inferior frontal gyrus (IFG, including Broca's area), paracentral lobe, and a small part of the dorsal visual cortex. The clusters in the STG, MTG, and IFG fall, to a great extent, within classically identified language regions (see white outline in *Figure 1A*; mapping language areas from *Fedorenko et al., 2010*). This major effect of deafness on individual differences in FC was uniquely strong for the AC. Replicating this analysis with multiple control regions (all atlas cortex areas not involved in audition or language; Harvard-Oxford Atlas) showed that AC-FC had a much more substantial change in variability due to deafness ($X^2$=2303.18, p<0.0001; *Appendix 1—figure 1*).

To determine which group has larger individual differences in these regions (*Figure 1B*), we computed the ratio of variability between the two groups (deaf/hearing) in the areas that showed a significant difference in variability (*Figure 1A*). The deaf show variability over twice as large as the

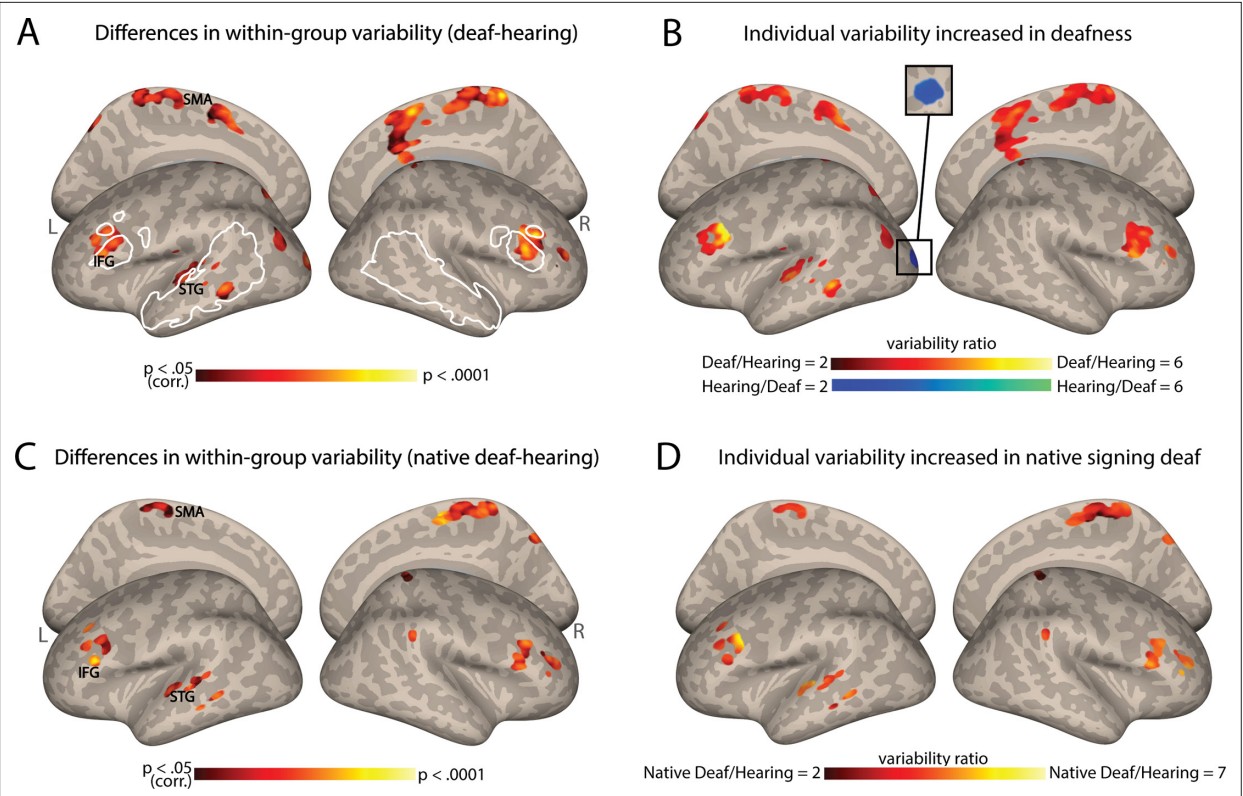

**Figure 1.** Individual differences in functional connectivity (FC) from the auditory cortex (AC) increase in deafness. (A) Significant differences in the interindividual variability of the AC-FC values between deaf and hearing groups (p<0.05, cluster-corrected for multiple comparisons) are presented on inflated cortical hemispheres. These are found in the left STG (including the auditory association cortex), bilateral IFG (including part of Broca's area), paracentral lobule, along with the dorsal visual stream. (B) The ratio of the within-group variability of AC-FC between the deaf and hearing groups is presented (within areas showing variability between the groups). Most areas showing a change in variability between the groups display larger individual differences in deafness, including the left auditory association cortex and Broca's area. (C) Differences in native signing deaf subgroup and hearing group in their interindividual variability of the AC-FC values (p<0.05, cluster-corrected for multiple comparisons) replicate the effect of the mixed deaf group (A). (D) The ratio of the variability of AC-FC between the native signing deaf and hearing (within areas showing variability difference between the groups). No area showed increased individual differences for the hearing group. Native signing deaf participants have higher individual differences, despite having no delay in language acquisition. Anatomical marks: SMA = supplementary motor area; IFG = inferior frontal gyrus; STG = superior temporal gyrus. The regions outlined in white show some of the language-sensitive regions identified by *Fedorenko et al., 2010*, including the IFG, the anterior and the posterior temporal parcellations.

The online version of this article includes the following figure supplement(s) for figure 1:

**Figure supplement 1.** Auditory cortex functional connectivity (AC-FC) variability difference between delayed signing deaf and hearing individuals.

hearing in most of the areas that show change to within-group variability, including the STG, MTG, and the IFG. The deaf group showed lower variability in only one cluster in the left early visual cortex. Thus, the findings from this analysis indicate that as in vision, in hearing individuals auditory experience appears to exert a general stabilizing influence on FC, whereas hearing loss leads to greater overall variability between individuals in the connectivity of the AC. A single exception is that the deaf had more consistent connectivity between their early auditory and visual cortices. This suggests that as in vision loss (*Sen et al., 2022*), individual differences dramatically increase due to deafness.

## Is the increased variability (mainly) explained by hearing loss?

Our sample of deaf individuals was rather homogenous in having severe to profound hearing loss from early life. However, it included a mix of native signers and adults who were deaf children to hearing parents, who were taught to sign later in life, and, in effect, experienced delayed language acquisition. Given that our sample of deaf individuals exhibited varying ages of language acquisition, it raises the question of whether the observed FC variability above is primarily attributable to delayed language acquisition or to hearing loss. To investigate this question, we tested if the increased variability would

**Table 1.** Participants' demographic information.

|  | Native deaf signers (N=16) | Delayed deaf signers (N=23) | Hearing nonsigners (N=33) |
|---|---|---|---|
| Age of sign language acquisition | 0±0 | 6.91±1.62 | N/A |
| Age | 28.50±7.13 | 27.09±5.87 | 21.97±2.54 |
| Years of education | 14.13±2.31 | 15.09±1.41 | 15.03±1.93 |
| Gender | 11 M, 5 F | 12 M, 11 F | 15 M, 18 F |

still be found when comparing native deaf signers to hearing individuals, all of whom had natural language experience (for sign or spoken language, respectively) from birth through their parents. Our results demonstrated a very similar pattern to the one described above, revealing increased variability in temporal, frontal, and medial regions (*Figure 1C*; see also *Table S1*). The FC variability is higher in the native signing deaf individuals when compared to the hearing individuals (*Figure 1D*). Similar findings are seen when comparing the deaf delayed language and hearing groups (e.g. for the IFG, *Figure 1—figure supplement 1*). This outcome suggests that deafness-related factors, even without delayed language acquisition, are sufficient to generate more diverse FC from the AC between individuals and that auditory experience, regardless of language exposure, exerts a broad stabilizing effect on FC.

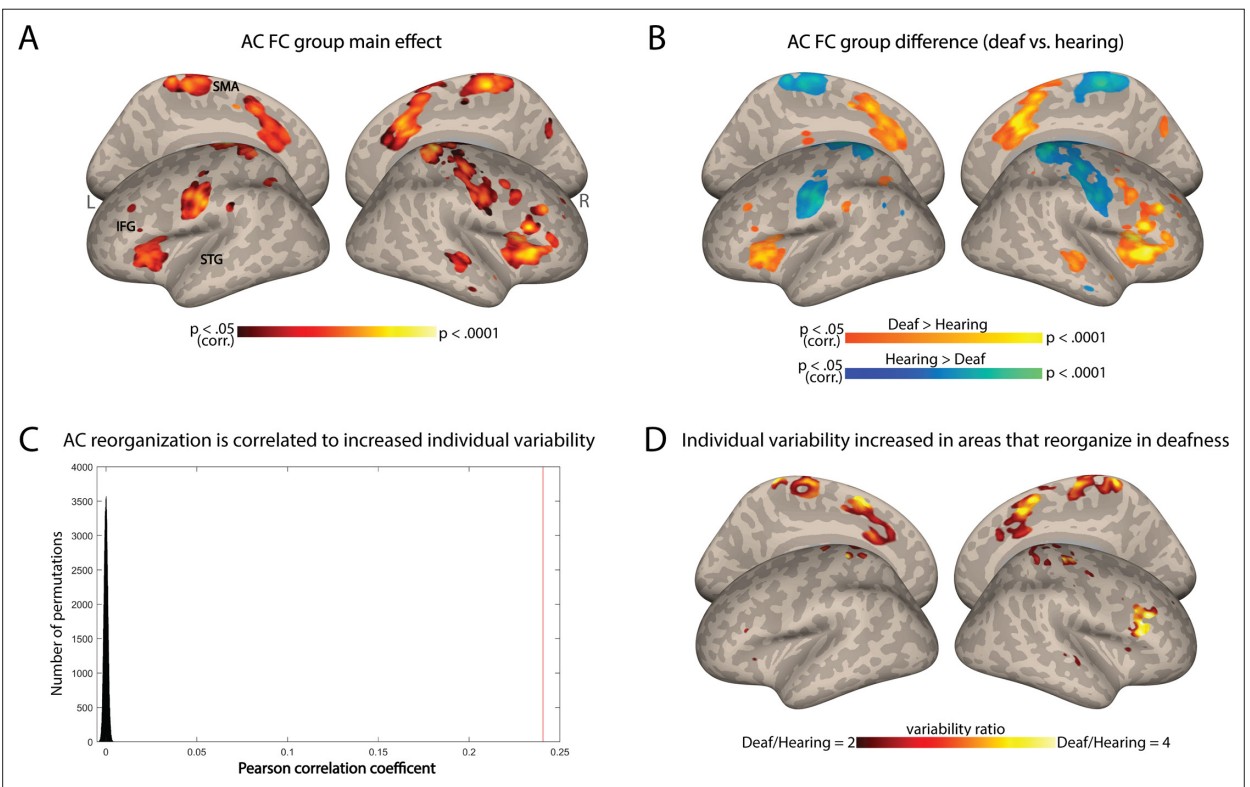

**Figure 2.** Individual variability in deafness is related to brain plasticity. (A) ANOVA main effect showing which regions are reorganized in deafness (group difference between the deaf and hearing in auditory cortex functional connectivity [AC-FC]) (p<0.05, cluster-corrected for multiple comparisons). (B) Direct comparison of AC-FC between deaf and hearing groups (p<0.05, cluster-corrected for multiple comparisons) broadly replicated previous findings, showing broad reorganization in deafness. (C) Correlation between regions that show increased individual differences (*Figure 1A*) with the regions that show reorganization in deafness (A) is shown as a red line (r=0.24) compared with a spatial permutation test (distribution in black); the brain patterns of FC reorganization and that of increased individual differences are correlated, suggesting increased individual differences characterizes plasticity in deafness. (D) The ratio of the intragroup variability of AC-FC between the deaf and hearing groups is shown within areas showing reorganization group-level changes to FC. No area showed increased individual differences for the hearing group. Among the areas showing a change in AC-FC in deafness, individual differences are overall increased (red-orange) or stable (uncolored). Anatomical marks: SMA = supplementary motor area; IFG = inferior frontal gyrus; STG = superior temporal gyrus.

## Does AC variability increase especially for areas that reorganize in deafness?

To test if this change in individual differences stems from variable outcomes of deafness-related plasticity, we tested if areas that show reorganization in FC are especially susceptible to increased individual differences. We computed the change in FC from the AC between the hearing and deaf groups (*Figure 2A*). Consistent with prior research (e.g. *Andin and Holmer, 2022*; *Ding et al., 2016*), deaf individuals showed increased FC to the AC in frontal, temporal, and parietal regions, while for the hearing the connectivity was stronger to sensorimotor areas (*Figure 2B*). We then explored whether regions that had undergone functional reorganization due to deafness also exhibited high variability within the deaf group. We predicted that if plasticity due to deafness results in higher variability, then areas with overall FC change between the groups would also display heightened variability within the deaf group, leading to a correlation between the two spatial maps. We therefore conducted a correlation analysis between the spatial pattern of variability difference observed between the groups (*Figure 1A*) and the spatial pattern of the group effect in terms of AC-FC (*Figure 2A*). The Pearson's correlation coefficient between these two maps was modest but highly significant (r=0.24, p<0.0001 for both smoothed and unsmoothed permuted data; confirmed through a permutation test shuffling voxel location across 100,000 iterations; *Figure 2C*). This suggests a moderate link between variability and plasticity: not only is the AC-FC more variable in the deaf, but the variability seems to also be increased in areas that showed reorganization because of deafness. To test which of the regions that had undergone reorganization had particularly variable plasticity across individuals, we inspected the variability ratio between the deaf and hearing groups in the areas that had group-level changes to FC. We found that all the areas that showed changes to FC exhibited either greater variability within the deaf group (mainly in the parietal and right frontal cortex) or similar variability in both groups (*Figure 2D*). No region showed higher variability in the hearing. Together, this suggests that plasticity FC of the AC in deafness is overall linked to more variable outcomes across individuals.

## Does delayed language acquisition affect individual differences?

Finally, we aimed to investigate the independent impact of language exposure, and whether delayed language acquisition played an additional role in the heightened variability observed among deaf individuals. To address this, we replicated the FC variability analysis by comparing deaf native signers to deaf delayed signers, equating hearing loss. In contrast to the results above, which revealed extensive variability change across multiple brain regions, this analysis only identified significant differences between native and delayed deaf signers in four small clusters (*Figure 3A*; see also *Table S1*) in the left posterior middle frontal gyrus (pMFG), close to the precentral gyrus (PreCG), the left posterior

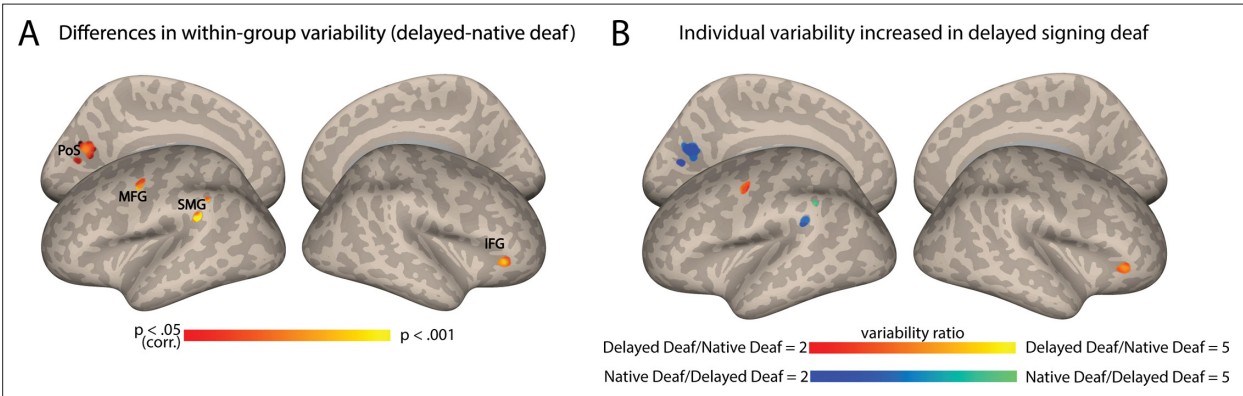

**Figure 3.** Auditory cortex functional connectivity (AC-FC) variability is influenced by language exposure. (A) Differences between delayed and native signing deaf subgroups in their interindividual variability of the AC-FC values (p<0.05, cluster-corrected for multiple comparisons) show changes to individual differences in four specific clusters. (B) The ratio of the variability of AC-FC between the delayed signing and native signing deaf (within areas showing variability between the subgroups) shows that individual differences increase due to delayed language acquisition in the left MFG and right anterior IFG, but that early-onset sign language exposure increases variability in the connectivity between the AC and the left SMG and left dorsal medial visual cortex (cuneus and precuneus). Anatomical marks: PoS = parietooccipital sulcus; IFG = inferior frontal gyrus; SMG = supramarginal gyrus; MFG = middle frontal gyrus.

supramarginal gyrus (pSMG), left dorsal visual cortex (precuneus and cuneus), and the right anterior inferior frontal gyrus (aIFG). Interestingly, these regions did not all show a consistent effect in their direction, but instead increased variability was attributed to both subgroups for different clusters. FC variability was increased for the deaf delayed signing individuals both in the left MFG and the right aIFG (*Figure 3B*). In contrast, the deaf native signing individuals showed higher variability in the pSMG and the dorsal stream (precuneus and cuneus) (*Figure 3B*). Interestingly, the two areas that had increased variability in delayed signers closely corresponded to language-related areas (e.g. pMFG and aIFG). These findings indicate that beyond the broader effects of deafness on individual differences in the FC of the early AC, delayed language acquisition can also affect individual differences, albeit to a lesser extent.

### Could other individual factors explain individual differences in deafness?

Our results so far suggest that the early lack of hearing experience (i.e. congenital deafness) is the primary factor driving AC-FC variability. However, even congenital deafness is not completely homogenous, and other factors related to partial hearing experience could also contribute to this individual variability among the deaf. For instance, the degree of hearing loss and the use of hearing aids, which provide residual hearing (even if not sufficient for language comprehension in the case of our participants), might also influence individual differences. To test this, we computed the correlation between AC-FC and three factors related to hearing experience: the age when hearing aids were first used, the duration of hearing aid use, and the hearing threshold. At the whole-brain level, we observed that AC-FC to different brain regions, primarily in the occipital lobe, but also in posterior MTG (*Appendix 1—figure 2*), are influenced by the age at which our deaf participants began using hearing aids, such that higher FC is correlated with older ages of onset. Interestingly, there was no correlation between the duration of hearing aid use and AC-FC with any brain region (no significant clusters, p<0.05, cluster-corrected for multiple comparisons). Last, we tested the correlation between AC-FC and hearing threshold. This analysis was possible only on a subset of our sample (N=23), with the remaining participants only able to report the level of hearing loss (i.e. 'profound') rather than specific hearing threshold values, and therefore should be interpreted cautiously. We found that AC-FC to the left fusiform gyrus is correlated with the hearing threshold of deaf participants (*Appendix 1—figure 2*), indicating that more profound hearing loss is associated with stronger FC between the AC and the fusiform gyrus.

Since these correlations did not implicate any of the areas we identified here as having higher AC-FC variability in the deaf (*Figure 1A and B*), we directly tested if the individual differences in AC-FC to these regions could be accounted for by these hearing-related parameters. We performed the same correlation analysis at the region of interest (ROI) level using individual clusters extracted from the map in *Figure 1A*. No significant correlations were found for any of the factors or ROIs (all p>0.05 before correction for multiple comparisons, see Appendix 1—figures 3-5 for scatterplots of AC-FC and each individual factor). These findings suggest that while hearing experience factors such as hearing threshold and age of hearing aid use can influence AC-FC, they do not account for the observed increased variability observed in our study, underscoring the role of congenital deafness in increasing AC-FC individual differences.

## Discussion

This study demonstrates a link between neural plasticity and variability in the AC of deaf individuals. Our study has demonstrated that, in comparison to a hearing group, individuals in the deaf group display a greater degree of individual variability in their FC from the AC. These were not driven by differences in their hearing threshold or hearing aid use experience, but rather by congenital auditory loss. This shows consistency with previous findings of increased individual differences in blindness (*Sen et al., 2022*). Furthermore, there is a mild relationship between this heightened variability and the adaptive changes occurring within the deprived AC. Specifically, we found that overall, the spatial patterns of plasticity and increased individual differences are significantly correlated, and there is increased variability in many areas functionally connected to the AC that have undergone reorganization due to deafness. Further, some, although more modest, increased variability was found when

comparing deaf individuals who had varying degrees of sign language acquisition, suggesting that language acquisition timing itself also plays an additional significant role in producing different AC functional connections. These findings suggest that although hearing loss in itself may be sufficient to increase individual differences, the variation of the FC patterns in the AC in response to deafness can increase further when considering a combination of auditory experience and timing of language acquisition. Together, these findings show how the interaction of auditory and language exposure may amplify the spectrum of FC diversity from the AC, allowing for a more comprehensive understanding of the complex factors shaping neural plasticity in response to deafness.

The auditory system, like the visual system, undergoes a critical phase during which its organization is fine-tuned by sensory experience (*Knudsen, 2004*). Given the fundamental role of auditory input for the AC organization, hearing loss leads to significant reorganization of this brain region. Importantly, while hearing loss induces cross-modal plasticity in the AC, we show that the extent of reorganization may vary among individuals with deafness. Recent evidence on blindness (*Sen et al., 2022*) suggests that the absence of visual experience increases individual differences in brain connectivity. Our study extends this inquiry to a different population experiencing a distinct type of sensory loss. Our findings demonstrate how hearing loss influences the neural connectivity profile of the deprived AC, introducing variability in the network outcomes. Much of the increased individual differences are found in areas that belong to the language system, including Broca's area (notably bilaterally) and the left STG and MTG (*Figure 1A and B*). This aligns with prior research demonstrating significant FC alterations between the AC and the language network in deaf infants and children (*Shi et al., 2016*; *Wang et al., 2019*). It also aligns with animal studies showing how deafness can impact top-down connectivity (*Yusuf et al., 2020*) and cortico-cortical interactions (*Yusuf et al., 2017*). Therefore, it appears that connectivity between the early AC and many language regions is stabilized or else affected by the use of audition for language in early life and becomes less consistent in its absence. Notably, while our findings mostly point to increased AC-FC variability within the deaf group, a single cluster in the left early visual cortex exhibits higher variability in the hearing group. This suggests a potential stabilizing impact of hearing loss on the interaction between the auditory and visual cortices, possibly due to the prevalent use of vision for adaptation. Overall, these results, coupled with those of *Sen et al., 2022*, highlight the impact of postnatal sensory experience in promoting consistency in brain organization, suggesting a general principle of brain development of the sensory systems.

The results of this study also provide evidence for the role of neural plasticity in generating diverse individual patterns of brain connectivity. Our finding that the exhibited heightened AC-FC variability by the deaf group corresponds spatially to regions that reorganize in deafness, even if only moderately (r=0.24), suggests that the increased variability may be partly attributed to the reorganization and adaptation of neural circuits in response to hearing loss. Although not all areas that changed their mean FC to the AC showed increased variability in the deaf group, we found higher AC-FC variability in regions such as the inferior frontal cortex and pre-supplementary motor area. These same brain regions exhibited functional reorganization in response to deafness, as illustrated in *Figure 2B*, consistent with prior resting-state fMRI research emphasizing functional changes following hearing loss (e.g. *Andin and Holmer, 2022*; *Ding et al., 2016*). Importantly, these changes in the mean connectivity of the AC likely reflect the change in its function in deafness. Studies have shown that the AC can be activated in deaf individuals when performing parallel visual tasks, indicating a shift in functional activation from auditory to visual processing (e.g. *Almeida et al., 2015*; *Benetti et al., 2017*; *Benetti et al., 2021*; *Bola et al., 2017*; *Bottari et al., 2014*; *Butler et al., 2017*; *Finney et al., 2001*; *Lomber et al., 2010*; *Meredith and Lomber, 2011*; *Petitto et al., 2016*; *Scott et al., 2014*), albeit typically for the same type of functional computation (*Cardin et al., 2020*; *Heimler et al., 2015*; *Lomber, 2017*; *Pascual-Leone and Hamilton, 2001*). Our results suggest that such functional responses may vary across individuals in accordance with their variable levels of FC alterations. Interestingly, in addition to finding increased variability in connectivity for areas that increase their connectivity in deafness, we also found higher AC-FC variability in regions that show a decreased FC to the temporal lobe in deafness, specifically the somatosensory cortex (e.g. *Andin and Holmer, 2022*; *Bonna et al., 2021*; *Ding et al., 2016*). Here too, previous research has identified differences in somatosensory involvement between deaf and hearing individuals, which has been linked to sign language and visual processing (*Bonna et al., 2021*; *Okada et al., 2016*). Therefore, it appears that any type of plasticity in the

connectivity of the AC, regardless of its direction (increased or decreased FC), may manifest variably across individuals.

What underlying mechanisms drive these changes, and what is their significance? Although the term 'reorganization' is debated, and some concerns exist about the functionality of neural reorganization (*Makin and Krakauer, 2023*), functional recruitment of AC for nonauditory tasks is supported by evidence from animal studies demonstrating causal links (*Lomber et al., 2010*; *Meredith and Lomber, 2011*; for a review, *Alencar et al., 2019*; *Lomber et al., 2020*). Such functional changes appear to arise from a combination of unmasking and re-weighting of nondominant inputs, rather than extensive anatomical alterations (*Alencar et al., 2019*; *Kral and Sharma, 2012*; *Kral and Sharma, 2023*; *Kupers et al., 2010*; *Pascual-Leone and Hamilton, 2001*). Our findings of increased variability in FC from the AC in the deaf suggest more than mere unmasking, as this variability cannot easily be explained without assuming some degree of reorganization.

Evidence for reorganization is found in research using congenitally deaf cats and other animal models, which have shown that auditory deprivation leads to significant disruptions in the development and maintenance of brain connectivity patterns. Although broadly anatomical connectivity pathways are maintained in deafness (*Butler et al., 2016*; *Butler et al., 2018*; *Kok and Lomber, 2017*), changes in connectivity are found, namely within and between sensory networks (*Sacco et al., 2024*). These changes map to the disruption of neural communication within and between the auditory system (*Kral et al., 2017*). For instance, the decoupling of supragranular and infragranular layers of cat's primary AC (*Yusuf et al., 2022*) and the disruption of top-down cortico-cortical connectivity in congenitally deaf cats (*Yusuf et al., 2020*) suggest that neural communication within and between cortical regions is highly sensitive to sensory input. These disruptions likely contribute to heterogeneity in how neural networks in individuals with hearing loss are reorganized. Anatomical changes may also help explain brain connectivity variability in deafness, as congenital auditory deprivation leads to reduced thickness in both primary and secondary auditory regions, particularly in deep layers (e.g. infragranular layers) (*Berger et al., 2017*). Additionally, auditory deprivation induces synaptic reduction in the same infragranular cortical layers, further disrupting the structural and functional integrity of these regions (*Kral et al., 2000*). Together, these findings align with the idea that anatomical and functional constraints interact with sensory experience to drive cortical plasticity, and that these may vary between individuals. Importantly, while group-level animal studies have provided valuable insights into the effects of deafness, they rarely address individual variability, an aspect that has been increasingly explored in human studies. In animal research, the typically small sample sizes make it even more likely that interindividual variability could obscure group-level findings or lead to null results. Investigating variability within individual animals could reveal important patterns or mechanisms masked by averaging across groups, offering a more nuanced understanding of deafness and its effects on brain reorganization.

Our study does not determine the origins of the individual differences observed, but we expect that they may arise from a blend of genetic and environmental factors. Auditory experiences seem to stabilize FC patterns, implying that experience-dependent pruning in the auditory system consolidates a consistent pattern of connectivity optimized for hearing. In its absence, AC connectivity may become more variable, depending on random and inherited individual differences in patterns of AC connectivity at birth. Further, diverse compensatory experiences of deaf individuals could enhance the variability of AC-FC. Our analysis showed that factors such as hearing threshold and the age at which hearing aids are first used can influence AC-FC. However, these factors do not fully account for the observed variability, suggesting that AC connectivity is susceptible to a complex interplay of environmental influences. Future studies involving deaf infants will be crucial to determining the balance in which AC-FC variability is driven by environmental influences and inherited connectivity patterns.

In addition to the effect of deafness itself, we have also demonstrated how a particular additional factor, namely language experience, may moderately affect variations in the FC of the AC. Although our data may be somewhat underpowered to fully explore this question, deaf individuals who have had exposure to sign language from birth, for example, appear to exhibit more consistent connectivity between the AC and the left pMFG, as well as the right aIFG, compared to those who had experienced delayed language acquisition in early development (*Figure 3B*). Interestingly, the left pMFG/PreCG have been associated with sign language comprehension (*Trettenbrein et al., 2021*; *Yang et al., 2024*), and our results indicate that delayed language acquisition leads to higher FC variability

in this area specifically. This provides some evidence that early experience with sign language consolidates this connectivity pattern. On the other hand, the aIFG, part of the inferior frontal cortex, is known to be involved in lexical comprehension and discourse semantics for sign language (*Emmorey, 2021*). The literature often highlights higher activation in the left hemisphere, typically accompanied by less extensive neural activity in the right hemisphere's homologous region. Therefore, it is unclear why the right aIFG shows higher FC variability in deaf delayed signers, as opposed to the left hemisphere; this may be due to variable reliance on areas capable of compensating for deficits in typical language systems (*Martin et al., 2023*; *Newport et al., 2022*; *Tuckute et al., 2022*). Further research is needed to elucidate the precise role of the right hemisphere's IFG in this context. In contrast, the AC connectivity to the left supramarginal gyrus and the left cuneus/precuneus is more consistent in people who experienced delayed language acquisition in addition to deafness. Although we only speculate why these areas show such interactions, these findings highlight the complex interplay between sensory experience, language acquisition, and neural plasticity in shaping the individual patterns of FC of the AC. However, these outcomes did not align with our initial hypothesis, which anticipated a more pronounced effect of increased individual differences in delayed signers, especially within the language network. This may be since the observed variations in brain connectivity from early AC are primarily attributed to hearing loss, rather than delayed language acquisition. In turn, this could be both due to the early AC function as primarily responsive to auditory stimulation and to the fairly early maturation of this region (*Kral and O'Donoghue, 2010*), which may make it more susceptible to hearing loss, rather than to delayed language acquisition itself. This conclusion is further reinforced by our analysis targeting the variability in deaf native signers: the findings showed similar patterns of increased individual variability for FC within this subgroup as compared to the hearing (*Figure 1C and D*) compared to the analysis involving both native and non-native signers (*Figure 1A and B*). It would appear that auditory loss in itself, regardless of language experience, is a larger driver of increased individual differences. Along the same lines, we have also tested if increased individual differences may then be found in the connectivity from Broca's area in the case of delayed language acquisition, and did not find any significant effect. Though this may be due to insufficient power, this further emphasizes that the increase in individual differences in AC-FC, and possibly beyond it, during deafness are primarily attributed to hearing loss. An additional variable that may contribute to the relatively minor effect of language experience in our results is the relatively high language abilities within our cohort of delayed signers, which were comparable to those of the native signers. All deaf participants self-reported consistent levels of sign language proficiency, a factor that is typically affected following delayed language acquisition (*Bogliotti et al., 2020*; *Caselli et al., 2021*; *Cheng and Mayberry, 2021*; *Tomaszewski et al., 2022*). Furthermore, a subset of delayed deaf signers acquired sign language before the age of 6 (N=6, see also *Supplementary file 2, table S2*), potentially rendering them less susceptible to the impact of language deprivation. To further elucidate these findings, future investigations should include a larger and more diverse sample, specifically in terms of sign language acquisition age, in order to comprehensively address this aspect.

Finally, hearing aids and cochlear implants represent the primary approaches in auditory rehabilitation, and individual differences could be considered with respect to these interventions. The effectiveness of these treatments, especially cochlear implantation, is intricately linked to the extent of reorganization within the AC (*Feng et al., 2018*; *Heimler et al., 2014*; *Heimler et al., 2015*; *Kral et al., 2019*; *Lee et al., 2001*; *Lee et al., 2007*). The ability to regain a lost sense (i.e. hearing) is likely influenced by the preservation of the auditory system, as cross-modal reorganization for a different function may hinder its capacity to process information from the original modality and computation. Although this link is nuanced, given that some portions of the AC appear to reorganize for parallel functions to those they typically perform (*Cardin et al., 2020*; *Heimler et al., 2015*; *Lomber, 2017*; *Pascual-Leone and Hamilton, 2001*), reorganization appears to affect the ability to restore auditory function to AC. Although future research would need to establish a direct link between the individual brain connectivity patterns reported here and their functional utility, the diverse reorganization levels shown in this study hold potential clinical relevance for auditory rehabilitation. This is particularly true when considering the larger individual differences in how strongly the AC connects to the language system (*Figure 1B*), where a disconnect may form between the reorganized role in visual language and auditory feed-forward roles. Additionally, this study highlights the imperative of acknowledging and considering differences between hearing and deaf individuals, particularly when employing normative

data in clinical contexts (e.g. neurosurgery). The recognition of variability in brain organization among diverse populations underscores the necessity for tailored approaches in clinical practices, ensuring more accurate and effective interventions for deaf individuals.

It is worth noting that we assessed individual differences based on FC during task performance and not at rest. Although it would be prudent for future research to explore this aspect, we expect that individual patterns of plasticity in the AC connectivity remain relatively consistent across different time periods and states. FC patterns of hearing individuals are primarily shaped by common system and stable individual features, and not by time, state, or task (*Finn et al., 2015*; *Gratton et al., 2018*; *Tavor et al., 2016*). While the task may impact FC variability, we have recently shown that individual FC patterns are stable across time and state even in the context of plasticity due to visual deprivation (*Amaral et al., 2024*). Therefore, we expect that in deafness as well there should not be meaningful differences between resting-state and task FC networks, in terms of FC individual differences.

In conclusion, this study demonstrates that the lack of auditory experience results in increased individual differences in brain organization. Notably, this increased variability is prominent in language areas and regions undergoing reorganization in response to deafness, highlighting the relationship between brain plasticity and individual differences. Furthermore, our findings indicate that this variability is not solely influenced by sensory loss due to deafness; deprivation from language during early life also plays a role in shaping this variability. Ultimately, these outcomes underscore the significance of postnatal experience in generating individual differences. Additionally, they support tailoring rehabilitation strategies to match the unique patterns of plasticity seen in individuals with sensory impairments, including those with deafness.

## Methods

### Participants

We recruited 39 congenitally or early deaf adults and 33 hearing college students, all native Mandarin Chinese speakers (15 males, mean age 21.97±2.58 years, range: 18–28 years; see *Table 1* for the detailed characteristics of the participants). All of them possessed normal or corrected-to-normal vision, and their majority was right-handed (with the exception of three deaf individuals), as determined by the Edinburgh inventory (*Oldfield, 1971*). Prior to their involvement in the study, all participants provided written informed consent and received monetary compensation for their participation. The research protocol was reviewed and approved by the Human Subject Review Committee at Peking University (2017-09-01), adhering to the principles outlined in the Declaration of Helsinki.

All participants with hearing impairment completed a background questionnaire, in which they provided information about their hearing loss conditions, history of language acquisition, and educational background *Supplementary file 2, table S2*. Specifically, the etiology of hearing loss was collected in a questionnaire, with the following four options: hereditary (selected by N=15), maternal disease (N=4), ototoxicity (N=9), and other/unknown (N=11 wrote 'unknown'). Note that we are unable to confirm the self-reported etiology of hearing loss due to the lack of medical records and the lack of systematic medical examinations for hearing loss in China 20–30 years ago. All deaf participants indicated severe (N=8) or profound (N=31) deafness from birth, except for three participants who reported becoming deaf before the age of 3. Self-reported hearing thresholds ranged from 85 to 120 decibels (dB). Some of the participants used hearing aids during their lifetime, however, speech comprehension was reported as very poor, even when hearing aids were employed. At the time of testing, five deaf participants were using hearing aids frequently (either daily or three to four times per week); one reported to have used hearing aids, only but rarely (one to two times per month); while others either had never used hearing aids or had used them for varying durations (with usage spanning from 0.5 to 20 years, see also *Supplementary file 2, table S2*). Only one deaf participant reported having received long-term oral training from teachers starting at age 2.

The deaf participants were divided into two distinct subgroups. The first subgroup, referred to as 'native signers', consisted of 16 individuals (11 males). These individuals were born to deaf parents and were exposed to Chinese Sign Language (CSL) shortly after birth. The second subgroup, known as 'delayed signers' (non-native signers), comprised of 23 individuals (12 males). These participants were born into hearing families and began learning CSL after enrolling in special education schools, with the age of CSL initiation ranging from 4 to 10 years. The two deaf groups were carefully matched

on various demographic variables, including gender, age, and years of education (p>0.15). Additionally, in terms of language skills, both deaf groups were matched in terms of self-reported proficiency in CSL comprehension, production, and lipreading skills (p>0.34).

The hearing group and the deaf group were matched based on gender and years of education (p>0.15), but there was a significant age difference between these two groups (p<0.05). Given this significant age difference, we used age as a nuisance variable in our FC analyses, and the differences in variability were assessed after statistically accounting for the age variable.

## Image acquisition

Functional and structural MRI data were collected using a Siemens Prisma 3T Scanner with a 64-channel head-neck coil at the Center for MRI Research, Peking University. Functional data were acquired with a simultaneous multi-slice echoplanar imaging sequence supplied by Siemens (62 axial slices, repetition time [TR]=2000 ms, echo time [TE]=30 ms, multiband factor = 2, flip angle [FA]=90°, field of view [FOV]=224 mm×224 mm, matrix size = 112×112, slice thickness = 2 mm, gap = 0.2 mm, and voxel size = 2 mm×2 mm×2.2 mm). A high-resolution 3D T1-weighted anatomical scan was acquired using the magnetization-prepared rapid acquisition gradient echo sequence (192 sagittal slices, TR = 2530 ms, TE = 2.98 ms, inversion time = 1100 ms, FA = 7°, FOV = 224 mm×256 mm, matrix size = 224 × 256, interpolated to 448×512, slice thickness = 1 mm, and voxel size = 0.5 mm×0.5 mm×1 mm).

## Image preprocessing

We used SPM12 (Welcome Trust Centre for Neuroimaging, London, UK), run in MATLAB R2018b (Mathworks, Inc, Sherborn, MA, USA), for processing and analysis of structural and functional data. For each participant, the first four volumes of each functional run were discarded for signal equilibrium. The remaining functional data were slice-time corrected to the first slice (middle slice in time) and corrected for head motion to the first volume of the first session using 7th degree B-spline interpolation. All participants had head motion less than 2 mm/2°, except for one hearing participant that showed excessive head motion in 2 runs, which were excluded from analysis. Structural images were coregistered to the first functional images. Functional data were then normalized to MNI anatomical space using a 12-parameter affine transformation model in DARTEL (*Ashburner, 2007*) and resampled to 2 mm$^3$ voxel size prior to applying a 6 mm FWHM Gaussian filter.

## Stimuli and procedure

During the fMRI scanning, the participants performed a semantic task whose predictors were regressed out to focus on the underlying FC patterns. Design-regressed task data have been extensively used in the past to calculate FC (e.g. *Amaral et al., 2021*; *Gratton et al., 2018*; *Norman-Haignere et al., 2012*; *Walbrin and Almeida, 2021*), and it has been shown that it effectively leads to similar FC estimates as when using resting scans (*Fair et al., 2007*). Stimuli comprised of a set of 90 written words. This set consisted of 40 concrete/object words and 50 abstract/nonobject words, the latter lacking explicit external referents. Participants were given instructions to visually examine each of these 90 target words, contemplate their meanings, and engage in an oddball one-back semantic judgment task (*Wang et al., 2023*).

Each participant completed a total of 10 runs of task fMRI scanning, with each run lasting for 360 s. One native signer completed only 8 runs and subsequently withdrew from the study due to discomfort, so we analyzed 8 runs for this subject. In each run, there were 90 target word trials, each lasting for 2.5 s, as well as 14 catch trials, also lasting 2.5 s each. For more details about this experiment, please see *Wang et al., 2023*. There was no difference in the activation for words across the brain (p<0.05, cluster-corrected for multiple comparisons) between the deaf and hearing participants in this task. Hearing and deaf participants performed differently in terms of accuracy ($acc_{deaf}$ = 74% vs. $acc_{hearing}$ = 89%, t(70) = 5.6, p<0.05), but not in terms of reaction time ($RT_{deaf}$ = 1083 ms vs. $RT_{hearing}$ = 1147 ms, t(70) = 1.3, p=0.2). Despite the groups being matched for reaction time, to further control for task performance effects, both task accuracy and reaction time were included as nuisance variables in all analyses (deaf vs. hearing; delayed deaf vs. native deaf).

## FC analysis

FC was computed using the CONN toolbox (*Whitfield-Gabrieli and Nieto-Castanon, 2012*). Time courses were extracted from the 10 runs after regressing out the task predictors, and potential confounding effects were estimated and removed separately for each voxel and for each participant and run. In addition, functional data were denoised using a standard denoising pipeline (*Nieto-Castanon, 2020*) including the regression of potential confounding effects characterized by white matter time series, CSF time series, motion parameters, session and task effects, and simultaneous bandpass frequency filtering of the BOLD time series (*Hallquist et al., 2013*) between 0.01 Hz and 0.1 Hz.

### Seed ROI

The seed region for the early AC was defined using the atlas provided by the CONN toolbox (Harvard-Oxford Atlas distributed with FSL; *Jenkinson et al., 2012*). We extracted the Heschl's gyrus parcellation (broadly corresponding to the location of the primary AC) for both hemispheres and used it as our seed region for the FC analysis.

### FC variability analysis

Seed-based connectivity maps for each subject were estimated characterizing the spatial pattern of FC with the seed area (bilateral Heschl's gyrus). FC strength was represented by Fisher-transformed bivariate correlation coefficients from a weighted general linear model, modeling the association between their BOLD signal time series. To examine whether there were differences in the interindividual variability of FC values between the two groups, namely the deaf and hearing participants, we conducted the Brown-Forsythe test for equal variance (*Figure 1A*). The Brown-Forsythe test (*Brown and Forsythe, 1974*) is a homogeneity of variance test like Levene's test, conventionally used to test for variability differences, but uses the median instead of the mean, safeguarding against false positives in cases of skewed data distribution (*Olejnik and Algina, 1987*). The regression of the age variable was implemented in the analyses comparing deaf vs. hearing (given the age difference between these groups), while the regression of task variables (i.e. accuracy and reaction times) was included in all analyses to account for task performance effects. The minimum significance level for all presented results was established at $p < 0.05$, cluster-corrected for multiple comparisons within the gray matter volume using the spatial extent method (a set-level statistical inference correction; *Forman et al., 1995*; *Friston et al., 1994*). Correction was based on the Monte Carlo simulation approach, extended to 3D datasets using the threshold size plug-in for BrainVoyager QX (Brain Innovation, Maastricht, Netherlands).

To inspect the direction of the variability group effect, and determine which group had higher variance, we computed the ratio of variability between the groups (variability deaf/variability hearing, *Figure 1B*; *Sen et al., 2022*) for each voxel showing a significant Brown-Forsythe test effect ($p < 0.05$, corrected). We also conducted equivalent analyses on a subset of the deaf participants, with our investigation centering on the roots of the differences in individual variability, and whether they stem from hearing loss (deafness) or from late exposure to language. To test the role of hearing loss, we compared deaf individuals who are native signers to hearing participants (*Figure 1C*), both populations having access to full language (spoken and CSL, respectively) from birth. To test the role of delayed language acquisition, we compared native signing deaf individuals to deaf individuals who acquired sign language at a later stage (*Figure 3*).

In addition to the variability analysis, FC data were also analyzed to directly compare the connectivity between the groups, with a one-way ANOVA (*Figure 2A*). To inspect the direction of reorganization in AC-FC, we computed a post hoc t-test comparing FC between the groups (deaf vs. hearing, *Figure 2B*). To quantitatively examine the link between reorganization in deaf individuals and its impact on variability, we conducted a comparative analysis between the spatial pattern of FC variability (*Figure 1A*) and the spatial pattern of reorganization observed in the deaf (*Figure 2A*). This was done with the unthresholded maps to correlate the spatial pattern at large between these statistical effects. This was achieved by calculating the Pearson's correlation coefficient between these maps, specifically within the gray matter (*Figure 2C*). The significance level for the correlation was obtained using a permutation test (100,000 iterations), randomly shuffling voxels for each iteration and convolving each random map with a Gaussian kernel based on data smoothness estimation to

account for spatial autocorrelation. To ensure the additional smoothing step does not introduce artificial correlations, we also calculated the significance level without applying Gaussian smoothing to the permuted maps. The resulting permutation distribution was then compared with the previously obtained Pearson's correlation coefficient. Finally, we also inspected the variability ratio within the areas that showed reorganization in deafness (*Figure 2D*).

## Acknowledgements

We are thankful to all subjects who participated in our experiment. We also thank Melody Schwenk for her comments on this work. This work was supported by the Edwin H Richard and Elisabeth Richard von Matsch Distinguished Professorship in Neurological Diseases (to ES-A); STI2030-Major Project 2021ZD0204100 (2021ZD0204104) (to YB); National Natural Science Foundation of China (31925020 to YB; 32171052 to XW).

## Additional information

### Competing interests

Yanchao Bi: Senior Editor, eLife. The other authors declare that no competing interests exist.

### Funding

| Funder | Grant reference number | Author |
|---|---|---|
| Edwin H. Richard and Elisabeth Richard von Matsch Distinguished Professorship in Neurological Diseases | | Ella Striem-Amit |
| STI2030-Major Project | 2021ZD0204104 | Yanchao Bi |
| National Natural Science Foundation of China | 31925020 | Yanchao Bi |
| National Natural Science Foundation of China | 32171052 | Xiaosha Wang |

The funders had no role in study design, data collection and interpretation, or the decision to submit the work for publication.

### Author contributions

Lenia Amaral, Conceptualization, Formal analysis, Investigation, Methodology, Writing - original draft; Xiaosha Wang, Data curation, Funding acquisition, Investigation, Writing – review and editing; Yanchao Bi, Supervision, Funding acquisition, Investigation, Writing – review and editing; Ella Striem-Amit, Conceptualization, Supervision, Funding acquisition, Investigation, Methodology, Writing – review and editing

### Author ORCIDs

Lenia Amaral ⓘ https://orcid.org/0000-0002-0631-7944
Xiaosha Wang ⓘ https://orcid.org/0000-0002-2133-8161
Yanchao Bi ⓘ https://orcid.org/0000-0002-0522-3372
Ella Striem-Amit ⓘ https://orcid.org/0000-0002-5900-3455

### Ethics

Prior to their involvement in the study, all participants provided written informed consent and received monetary compensation for their participation. The research protocol was reviewed and approved by the Human Subject Review Committee at Peking University (2017-09-01), adhering to the principles outlined in the Declaration of Helsinki.

Reviewer #1 (Public review): https://doi.org/10.7554/eLife.96944.4.sa1
Reviewer #3 (Public review): https://doi.org/10.7554/eLife.96944.4.sa2

Author response https://doi.org/10.7554/eLife.96944.4.sa3

## Additional files

### Supplementary files

Supplementary file 1. MNI coordinates (x, y, z) for the functional connectivity (FC) variability analyses.

Supplementary file 2. Additional characteristics of deaf participants.

MDAR checklist

### Data availability

Seed-based connectivity maps, code, and the source maps for the figures, have been deposited in OSF at the link https://doi.org/10.17605/OSF.IO/YVKMF. Deidentified raw nifti files are not publicly available due to ethical constraints, however, they are available from the corresponding authors upon reasonable request. Researchers who are interested in the raw neuroimaging data for scientific purposes could contact the corresponding author (ybi@bnu.edu.cn) by submitting a brief project proposal. The proposal will be further assessed by the Human Subject Review Committee at Peking University. Any commercial research is not allowed to be performed on the data.

The following previously published dataset was used:

| Author(s) | Year | Dataset title | Dataset URL | Database and Identifier |
| --- | --- | --- | --- | --- |
| Amaral L, Wang X, Bi Y, Striem-Amit E | 2024 | Unraveling the impact of congenital deafness on individual brain organization | https://doi.org/10.17605/OSF.IO/YVKMF | Open Science Framework, 10.17605/OSF.IO/YVKMF |

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

## Appendix

### Functional connectivity variability analysis

This analysis was replicated for additional control ROIs – including all ROIs from the Harvard-Oxford Atlas not related to audition or language, and the number of significant voxels was calculated per ROI to test if the change in variability in deafness is specific to the AC. To determine if the number of voxels in AC was significantly higher than in the other control ROIs, we conducted a chi-square test for goodness of fit (*Appendix 1—figure 1*).

### Correlation with hearing experience variables

In order to inspect the effect of specific factors related to hearing experience (see also *Supplementary file 2, table S2*) on AC-FC variability, we calculated the correlation between the AC-FC of each voxel for deaf participants and: (1) hearing threshold, defined as the lowest dB across both ears (numeric value available for 23 participants); (2) hearing aid use start age, for 28 participants who reported having used hearing aids; (3) hearing aid use, for all deaf participants, except one who did not report the exact length of use for the hearing aids (N = 38). These correlations were computed at the whole-brain level (*Appendix 1—figure 2*; using the same multiple-comparisons correction as previously mentioned). Further, they were computed at the ROI level (*Appendix 1—figures 3–5*) for all the clusters that showed a main effect of increased AC-FC variability in deafness (*Figure 1A*). None of these correlations were significant even before correction for multiple comparisons (all p > .05).

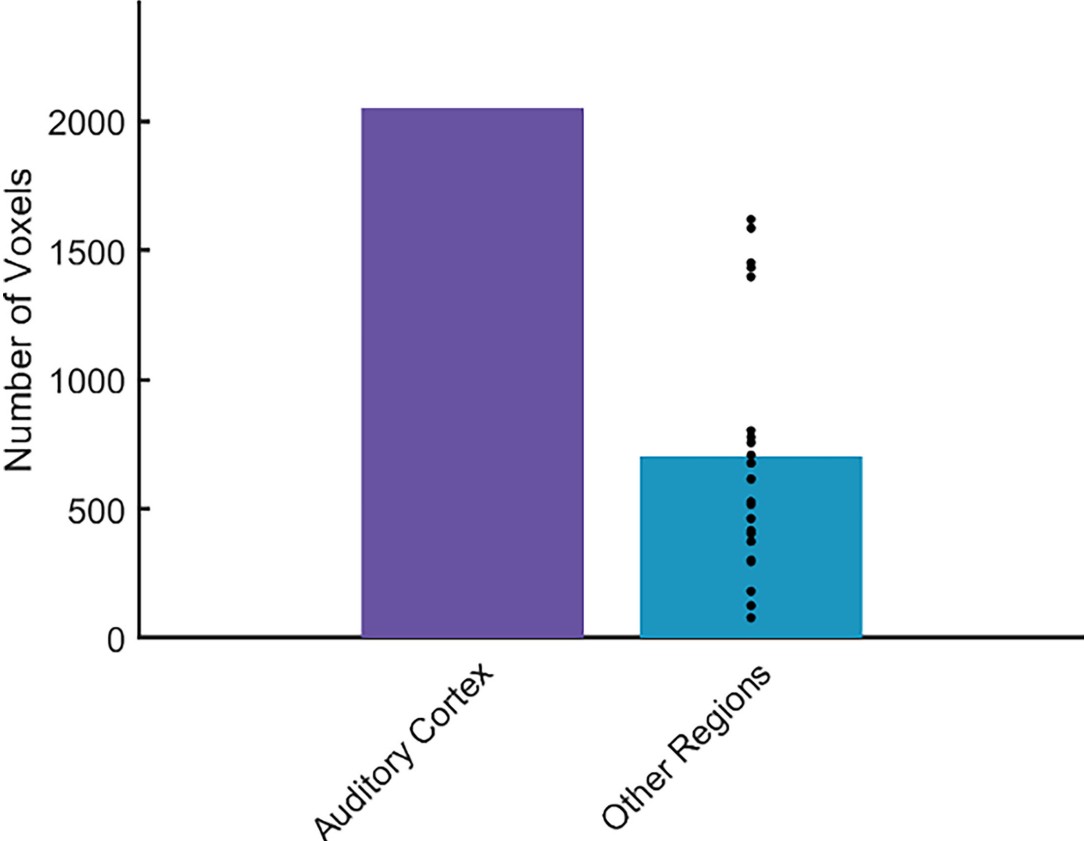

**Appendix 1—figure 1.** To test if increased variability in deafness was unique to the auditory system, we calculated inter-individual variability using all regions from the Harvard-Oxford Atlas, excluding auditory and language regions, as control seed regions for the functional connectivity (FC) analysis. FC variability from the seed areas to the whole brain was calculated for the Auditory Cortex (AC) and control regions. The plot shows the number of voxels with a significant Brown-Forsythe effect (change in variability) for the AC region (purple bar) and the average number of significant voxels using the control regions (blue bar). The black dots represent the number of significant voxels for each individual control region. The connectivity variability from the AC is uniquely increased compared to these control regions ($X^2$ = 2303.18, p < .0001).

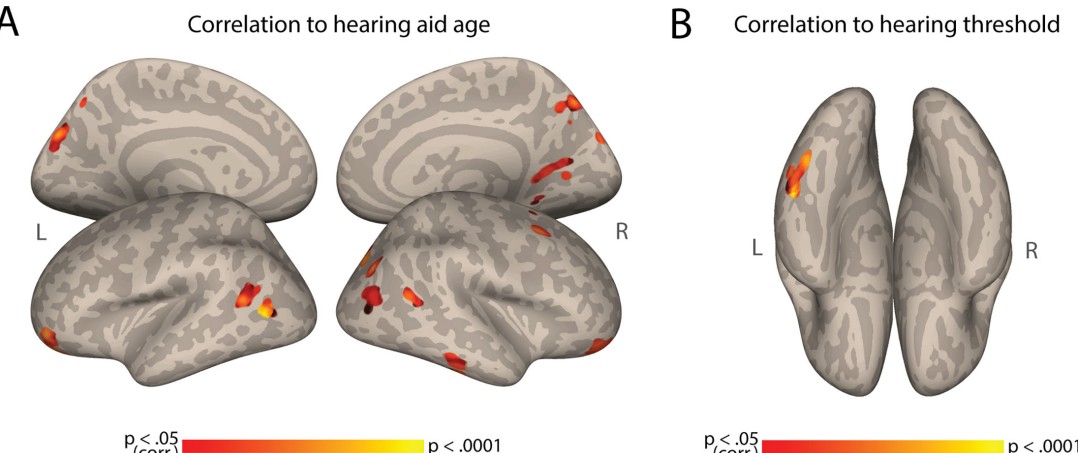

**Appendix 1—figure 2.** Correlation to hearing aid age and hearing threshold. (**A**) Auditory Cortex Functional Connectivity (AC-FC) to different regions is positively correlated to the age of deaf participants started using hearing aids (N=28). (**B**) AC-FC to the fusiform gyrus is positively correlated to the hearing threshold of deaf participants (N = 23). Both maps are cluster-corrected for multiple comparisons, p < .05.

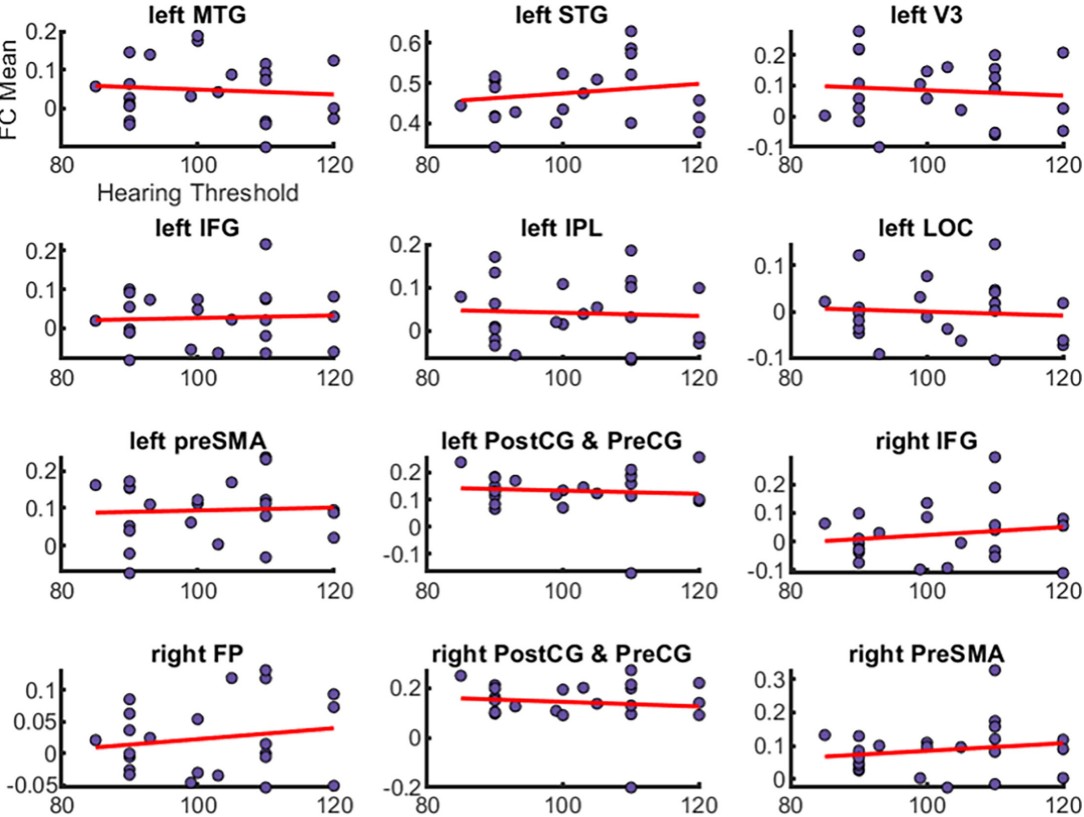

**Appendix 1—figure 3.** Scatter plots depicting the correlation between auditory cortex functional connectivity (AC-FC) mean values and hearing thresholds across various brain regions in deaf participants. The x-axis represents the hearing threshold (in decibels), while the y-axis represents the AC-FC mean value for the cluster. Each purple dot indicates individual participant data points, and the red line represents the trend line (linear regression) for each region. No significant correlations were found between FC mean and hearing threshold (all p > .05). MTG = middle temporal gyrus, STG = superior temporal gyrus, V3 = visual area 3, IFG = inferior frontal gyrus, IPL = inferior parietal lobe, LOC = lateral occipital cortex, preSMA = pre supplementary motor area, postCG = postcentral gyrus, preCG = precental gyrus, FP = frontal pole.

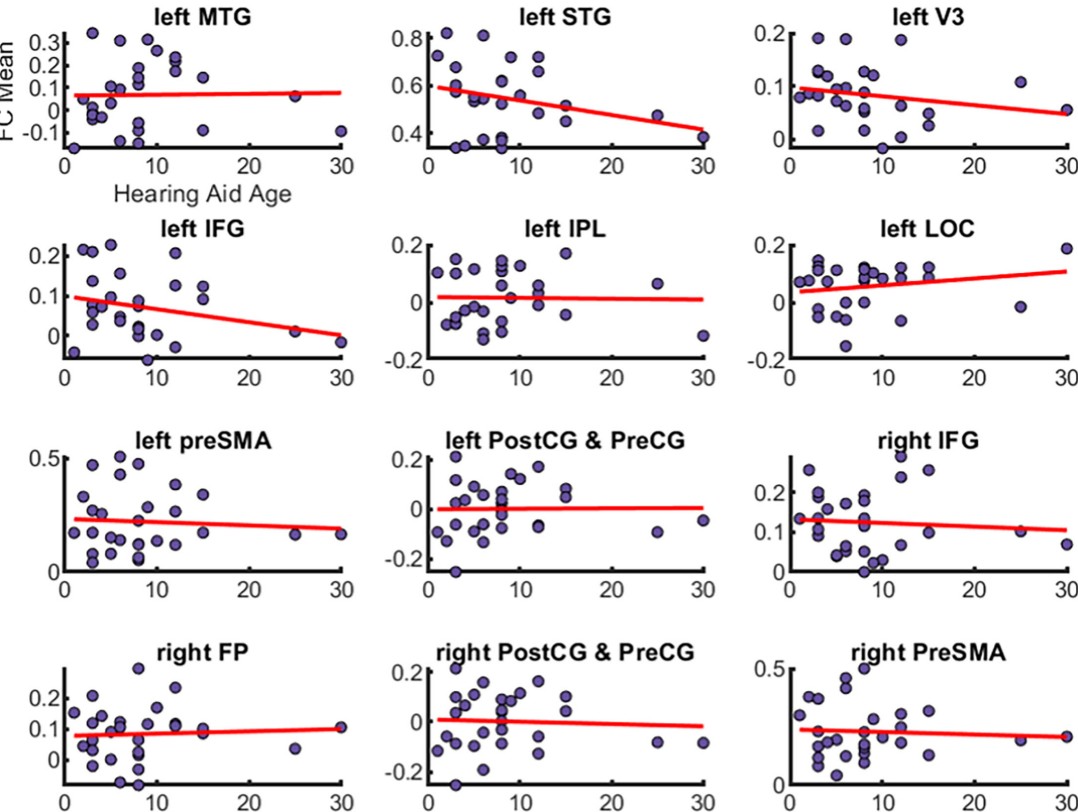

**Appendix 1—figure 4.** Scatter plots depicting the correlation between auditory cortex functional connectivity (AC-FC) mean values and age at which hearing aids were first used across various brain regions in deaf participants. The x-axis represents the hearing aids age in years while the y-axis represents the AC-FC mean value for the cluster. Each purple dot indicates individual participant data points, and the red line represents the trend line (linear regression) for each region. No significant correlations were found between FC mean and hearing aids age (all p > .05). MTG = middle temporal gyrus, STG = superior temporal gyrus, V3 = visual area 3, IFG = inferior frontal gyrus, IPL = inferior parietal lobe, LOC = lateral occipital cortex, preSMA = pre supplementary motor area, postCG = postcentral gyrus, preCG = precental gyrus, FP = frontal pole.

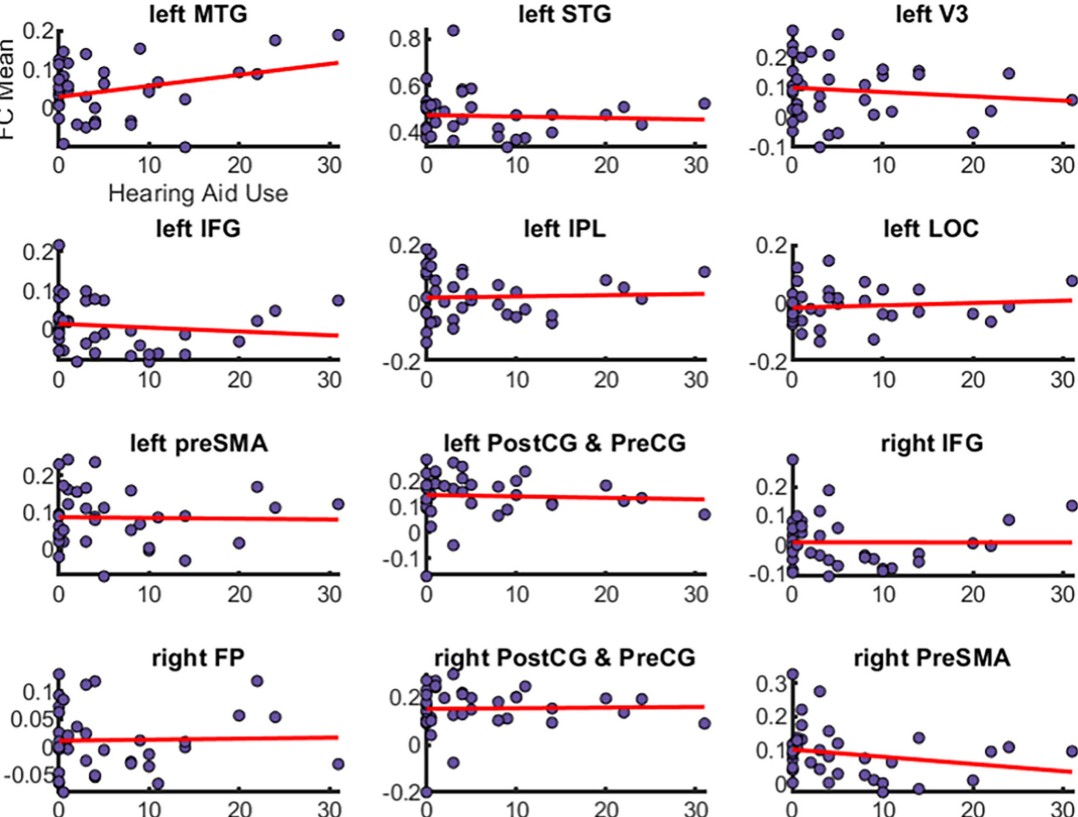

**Appendix 1—figure 5.** Scatter plots depicting the correlation between auditory cortex functional connectivity (AC-FC) mean values and duration of hearing aids use across various brain regions in deaf participants. The x-axis represents the duration of hearing aids use (in years), while the y-axis represents the AC-FC mean value for the cluster. Each purple dot indicates individual participant data points, and the red line represents the trend line (linear regression) for each region. No significant correlations were found between FC mean and hearing aids use (all p > .05). MTG = middle temporal gyrus, STG = superior temporal gyrus, V3 = visual area 3, IFG = inferior frontal gyrus, IPL = inferior parietal lobe, LOC = lateral occipital cortex, preSMA = pre supplementary motor area, postCG = postcentral gyrus, preCG = precental gyrus, FP = frontal pole.

