## [Editor Report · eLife Assessment]

This study presents **valuable** data on the increase in individual differences in functional connectivity with the auditory cortex in individuals with congenital/early-onset hearing loss compared to individuals with normal hearing. The evidence supporting the study's claims is **convincing**, although additional work using resting-state functional connectivity in the future could further strengthen the results. The work will be of interest to neuroscientists working on brain plasticity and may have implications for the design of interventions and compensatory strategies.

---

## [Referee Report · Reviewer #1 (Public review)]

This experiment sought to determine what effect congenital/early-onset hearing loss (and associated delay in language onset) has on the degree of inter-individual variability in functional connectivity to the auditory cortex. Looking at differences in variability rather than group differences in mean connectivity itself represents an interesting addition to the existing literature. The sample of deaf individuals was large, and quite homogeneous in terms of age of hearing loss onset, which are considerable strengths of the work. The experiment appears well conducted and the results are certainly of interest.

Comment from Reviewing Editor: In the revised manuscript, the authors have addressed all concerns previously identified by reviewer 1.

---

## [Referee Report · Reviewer #3 (Public review)]

Summary:

This study focuses on changes in brain organization associated with congenital deafness. The authors investigate differences in functional connectivity (FC) and differences in the variability of FC. By comparing congenitally deaf individuals to individuals with normal hearing, and by further separating congenitally deaf individuals into groups of early and late signers, the authors can distinguish between changes in FC due to auditory deprivation and changes in FC due to late language acquisition. They find larger FC variability in deaf than normal-hearing individuals in temporal, frontal, parietal, and midline brain structures, and that FC variability is largely driven by auditory deprivation. They suggest that the regions that show a greater FC difference between groups also show greater FC variability.

Strengths:

The manuscript is well-written, and the methods are clearly described and appropriate. Including the three different groups enables the critical contrasts distinguishing between different causes of FC variability changes. The results are interesting and novel.

Weaknesses:

Analyses were conducted for task-based data rather than resting-state data. The authors report behavioral differences between groups and include behavioral performance as a nuisance regressor in their analysis. This is a good approach to account for behavioral task differences, given the data. Nevertheless, additional work using resting-state functional connectivity could remove the potential confound fully.

Comment from Reviewing Editor: In the revised manuscript, the authors have addressed all concerns previously identified by reviewer 3, and the eLife assessment statement reflects the point by reviewer 3 that using resting-state functional connectivity in the future could further strengthen the results.

---

## [Author Response]

The following is the authors’ response to the previous reviews.

**Public Reviews:**

**Reviewer #1 (Public review):**
This experiment sought to determine what effect congenital/early-onset hearing loss (and associated delay in language onset) has on the degree of inter-individual variability in functional connectivity to the auditory cortex. Looking at differences in variability rather than group differences in mean connectivity itself represents an interesting addition to the existing literature. The sample of deaf individuals was large, and quite homogeneous in terms of age of hearing loss onset, which are considerable strengths of the work. The experiment appears well conducted and the results are certainly of interest. R: Thank you for your positive and thoughtful feedback.
**Reviewer #3 (Public review):**
Summary:This study focuses on changes in brain organization associated with congenital deafness. The authors investigate differences in functional connectivity (FC) and differences in the variability of FC. By comparing congenitally deaf individuals to individuals with normal hearing, and by further separating congenitally deaf individuals into groups of early and late signers, the authors can distinguish between changes in FC due to auditory deprivation and changes in FC due to late language acquisition. They find larger FC variability in deaf than normal-hearing individuals in temporal, frontal, parietal, and midline brain structures, and that FC variability is largely driven by auditory deprivation. They suggest that the regions that show a greater FC difference between groups also show greater FC variability.Strengths:The manuscript is well-written, and the methods are clearly described and appropriate. Including the three different groups enables the critical contrasts distinguishing between different causes of FC variability changes. The results are interesting and novel.Weaknesses:Analyses were conducted for task-based data rather than resting-state data. The authors report behavioral differences between groups and include behavioral performance as a nuisance regressor in their analysis. This is a good approach to account for behavioral task differences, given the data. Nevertheless, additional work using resting-state functional connectivity could remove the potential confound fully.The authors have addressed my concerns well.

Thank you for your thoughtful feedback. We appreciate your acknowledgment of the strengths of our study and the approaches taken to address potential confounds. As noted, we discuss the limitation of not including resting-state data in the manuscript, and we agree that this represents an important avenue for future research. We hope to address this question in future studies.